# Skin Lesion Classification Using Densely Connected Convolutional Networks with Attention Residual Learning

**DOI:** 10.3390/s20247080

**Published:** 2020-12-10

**Authors:** Jing Wu, Wei Hu, Yuan Wen, Wenli Tu, Xiaoming Liu

**Affiliations:** 1Hubei Province Key Laboratory of Intelligent Information Processing and Real-Time Industrial System, College of Computer Science and Technology, Wuhan University of Science and Technology, Wuhan 430081, China; huwei@wust.edu.cn (W.H.); toto10241024@163.com (W.T.); lxmspace@gmail.com (X.L.); 2School of Computer Science and Statistics, Trinity College Dublin, Dublin 2, Ireland

**Keywords:** attention, dermoscopy, residual learning, densely connected convolutional networks

## Abstract

Skin lesion classification is an effective approach aided by computer vision for the diagnosis of skin cancer. Though deep learning models presented advantages over traditional methods and brought tremendous breakthroughs, a precise diagnosis is still challenging because of the intra-class variation and inter-class similarity caused by the diversity of imaging methods and clinicopathology. In this paper, we propose a densely connected convolutional network with an attention and residual learning (ARDT-DenseNet) method for skin lesion classification. Each ARDT block consists of dense blocks, transition blocks and attention and residual modules. Compared to a residual network with the same number of convolutional layers, the size of the parameters of the densely connected network proposed in this paper has been reduced by half, while the accuracy of skin lesion classification is preserved. Our improved densely connected network adds an attention mechanism and residual learning after each dense block and transition block without introducing additional parameters. We evaluate the ARDT-DenseNet model with the ISIC 2016 and ISIC 2017 datasets. Our method achieves an ACC of 85.7% and an AUC of 83.7% in skin lesion classification with ISIC 2016 and an average AUC of 91.8% in skin lesion classification with ISIC 2017. The experimental results show that the method proposed in this paper has achieved a significant improvement in skin lesion classification, which is superior to that of the state-of-the-art method.

## 1. Introduction

Melanoma is one of the deadliest skin cancers in the world [1]. It develops from the pigment-producing cell and typically occurs in the skin but is rarely found in other parts of the human body, such as the mouth, nose and eyes. Malignant melanoma can evolve congenitally, be acquired from benign melanocytic nevus, evolve malignantly from a dysplastic nevus or be new growth. In recent years, the incidence and mortality of malignant melanoma have increased annually, and the age of death is much younger compared with other types of cancers. Though melanoma is a deadly disease, it has a high probability of being cured if it is found and detected early [2].

Dermoscopy [3] is an effective inspection method to detect skin lesions. By eliminating the reflection on the skin surface and optically amplifying a specific area by many times, it is possible to observe the skin underneath the epidermis and the junction of the real epidermis and the upper dermis, which are structures usually invisible to the naked eye.

Dermoscopy for the observation of pigmented skin lesions, vascular diseases and inflammatory diseases, for the auxiliary diagnosis and differential diagnosis of benign and malignant tumors of the skin and for non-invasive skin monitoring and remote medical treatment avoids unnecessary skin biopsies to a certain extent. It provides patients with a new, non-invasive skin detection method and greatly reduces the pain from an invasive examination. It is also capable of assisting symptom diagnosis and helping doctors to evaluate the disease and analyze the cause.

In practice, the accurate classification of skin lesions in dermoscopy images remains as a challenging problem due to the large scale of intra-class variation and inter-class similarity caused by the differences in imaging modalities and clinical pathologies and the limited ability to focus on semantic regions [4,5].

Deep learning-based methods for skin lesion classification have been broadly accepted in recent years. Convolutional neural networks (CNN) have brought a series of breakthroughs to image classification. The depth of the network is one of the critical factors in the design of the CNN architecture. In general, we usually increase the depth of the network and then expect a better feature expression capability and an enhanced prediction accuracy.

A deep convolutional neural network (DCNN) trained for classification has significant localization ability and can highlight the image recognition area [6]. However, the deepening of the network will cause gradient vanishing or exploding, which hinders the convergence of DCNN and its training process. In recent research, this problem has been solved by normalization initialization [7,8,9], which enables networks with dozens of layers to begin to converge to random gradients using stochastic gradient descent (SGD) [10]. He et al. [11] introduced a deep residual learning framework to solve the degradation problem, making it possible to build very deep residual networks, such as AlexNet [12], VGGNet [13], GoogLeNet [14], Inception [15] and so on. Later on, DenseNet [16] proposed the idea of densely connected layers, wherein each layer can obtain the connection feature map of the previous layer. These two main ideas are applied to different application scenarios and network architectures [17,18,19,20].

The attention mechanism stems from the study of human vision. Humans selectively focus on certain visible information of interest and ignore what they feel is irrelevant. This allocates available processing resources to the most informative part of the input signal. Typically, it is implemented in combination with gated functions (such as softmax or sigmoid) technology. In recent years, attention mechanisms have been widely used in deep learning applications, such as image and video captioning [21], Visual Question Answering (VQA) [22], machine translation [23] and image analysis [24,25,26,27,28]. One study [28] proposed an VGGNet-based method combined with an attention module. The attention map is estimated by the attention module, which highlights the image area of interest related to the classification of the skin lesion. There are many other methods that adopt attention mechanisms to classify skin lesions [23,29]. In image classification, although the attention mechanism effectively improves the performance of the model, the attention weights in these models are learned through a learnable layer with a large number of additional parameters, which leads to increased computational cost.

To address these problems, we propose a skin lesion classification model based on an attention mechanism and the residual learning of densely connected convolutional neural networks. One study [30] proposed a densely connected residual block to ensure the deep supervision of the convolutional neural network, effective gradient flow and feature reuse capabilities. Another study [31] proposed an attention model for skin lesion classification without introducing additional parameters. Inspired by these studies [30,31], the improved densely connected network added the attention and residual learning after each dense block and transition block. The proposed method can reduce the number of parameters and calculation operations, while achieving feasible accuracy without using any special hardware-software equipment to reduce the cost of training and inference processes. Joining residual learning helps to reverse the gradient information flow of the deep network and to strengthen the deep supervision of the network. The attention module learned together with other network parameters can adaptively focus on the distinguishing parts of skin lesions, highlighting the image regions of interest related to skin lesion classification. It uses the inherent self-attention ability of the convolutional neural network and uses the feature map obtained by the high layer as the low attention mask, rather than learning attention masks with extra layers. We evaluated our method with the ISIC 2016 [32] and ISIC 2017 datasets [33]. There are two binary classification tasks: one is to classify the lesion type as melanoma or other and one is to classify it as seborrheic keratosis or other. The experimental results show that the method proposed in this paper yields state-of-the-art classification performance in the classification of skin lesions.

This paper makes the following contributions:We propose a new densely connected convolutional network with an attention and residual learning (ARDT-DenseNet) model to precisely classify skin lesions in dermoscopy images. The model consists of multiple ARDT blocks, a global average pooling layer and a classification layer. Each ARDT block is combined with dense blocks, transition blocks and an attention residual learning module. Residual learning prevents the gradient from vanishing when the network becomes deeper. The attention map automatically highlights the image area related to the classification, with additional interpretable information over a class label.Compared to the hundreds of layers of the residual network, the number of network model parameters with hundreds of layers of densely connected networks is less than half, reducing the computing costs while the accuracy of skin lesion recognition is comparable.The proposed model is an end-to-end network architecture. Our method has achieved state-of-the art performance for skin lesion classification in the ISIC 2016 and ISIC 2017 datasets.Next, the design and architecture of our ARDT-DenseNet model will be introduced in detail in Section 2. Section 3 is the experimental process and results, and Section 4 is the discussion and conclusions. Finally, Section 5 is about future work.

## 2. Materials and Methods

### 2.1. Network Architecture

The proposed ARDT-DenseNet model consists of multiple ARDT blocks, a global average pooling layer and a classification layer. Each ARDT block is a combination of dense blocks, transition blocks and attention and residual modules. The proposed architecture in this paper is shown in Figure 1, and the inner details of the dense blocks and transition blocks are shown in Figure 2.

The depth of DenseNet100 30 is 100 layers, which has nearly half of the model parameters of ResNet101. We adopt DenseNet100 with an improved first convolutional layer. In the original DenseNet, Conv1 has a convolutional kernel with dimensions of 7 × 7 and stride of 2 and dimensions of 1 × 1 and 2 × 2 in the transform block. The transition block was mainly composed of a convolutional layer and an average pooling layer, and we changed the output channel of the transition block to be consistent with the input channel of the current ARDT block. It is worth noting that each convolutional operation inspired by Reference [31] uses a pre-activation method, which first includes regularization processing, then activation and finally convolution.

In order to adapt to the skin lesion classification and expand the receptive field of the convolutional operation to obtain more information, the Conv1 was replaced with a 3 × 3 convolution, and the convolution layer had 4*k* (*k* is growth rate) convolution kernels of size 3 × 3 with padding and a stride of 1. Each dense block has 12 layers of 1 × 1 and 3 × 3 convolutions. The growth rate of the feature map following each layer is set to 12, and there are four dense blocks in total, as shown in Table 1.

As shown in Figure 2, the transition blocks are composed with a 1 × 1 convolutional layer and a 2 × 2 average pooling layer. The 1 × 1 convolution in each transition block comes with a 4 × 4 convolutional kernel, which, in order to adapt to the recognition of skin lesions, increases the receptive field of the convolution process and obtains more information. After four dense blocks, a global average pooling layer is followed. Since the purpose of the paper is to deal with the binary classification of skin lesions (melanoma vs. others and seborrheic keratosis vs. others.), this paper uses a fully connected layer softmax function that has 2 neurons. Moreover, our ARDT-DenseNet model reduces vanishing-gradient and strengthens the delivery of the feature, and the feature is used more effectively. At the same time, the number of parameters is reduced to a certain extent.

It is worth mentioning that the convolutional network generally adopts the order of convolutional operation, batch regularization and activation functions, while the convolutional operation in the dense blocks and transition blocks here uses the way of pre-activation, with batch regularization first, then the activation function and finally the convolutional operation. In the model proposed in this paper, each dense block uses 12 layers. There is a total number of five dense blocks and one hundred convolutional layers in depth.

The degradation is not caused by overfitting, and adding more layers to a suitable depth model will lead to higher training errors. We introduced residual learning, as shown in Figure 3, to avoid this problem. Rather than introducing a new convolutional layer, residual learning is added after each dense block and transition block. Adding residual learning increases the information transmission of the effective gradient stream, which is more conducive to the training of deep networks. Since the resolution of the image is doubled after the transition block, the corresponding residual learning uses a 2 × 2 average pooling layer, which reduces the resolution of the original image by half.

Not all of the regions in the image have equal significance to the classification. Only those areas related to the task need to be concerned and to gain attention from the network. This learning mechanism enhances the propagation of gradient information, is conducive to the training of deep network models and strengthens the networks attention to the critical semantic information of lesion images. The attention mechanism usually uses additional trainable layers to learn attention weights. Inspired by Reference [31], the attention mechanism uses the inherent self-attention ability of DCNNs. This not only reduces the number of parameters of the network but also makes the network pay more attention to the semantically important fields and speeds up the training of the network. Finally, the residual learning is combined with the self-attention learning mechanism to improve the ability of network identification, enhance the transmission of gradient information in the process of network training and realize deep supervision. We chose the spatial attention mechanism in the paper to generate an attention mask from a high-level attention map. The specific structure of the introduced attention and residual learning is shown in Figure 3. It is worth noting not only that our attention module does not introduce additional parameters but also that our overall structure reduces parameters compared with Reference [30].

**Figure 3 sensors-20-07080-f003:**
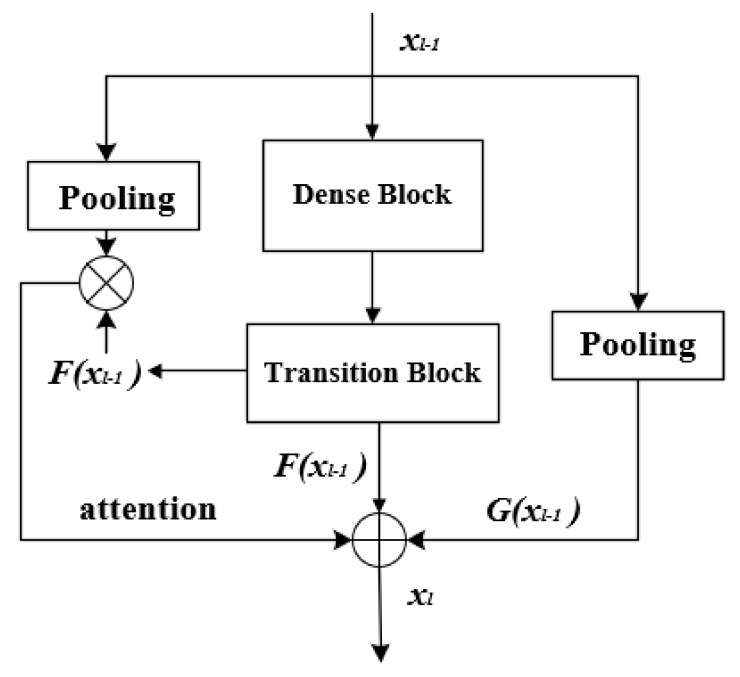
Architecture of ARDT Blocks. The “⊗ “represents matrix multiplication.

Suppose the input of the ARDT Block is recorded as $x_{l-1}$, the result of the residual learning output is recorded as *G*$\left(x_{l-1}\right)$ and the outputs of the dense block and the transition block are recorded as *F*$\left(x_{l-1}\right)$.

The normalization function is shown as Equation (1):(1)$$N^{sp}\left(F\left(x_{l-1}\right)\right)\;=\frac{e^{F{\left(x_{l-1}\right)}_{i,j}^{c}}}{\sum_{i^{\prime }j^{\prime }}e^{F{\left(x_{l-1}\right)}_{i^{\prime }j^{\prime }}^{c}}}$$
where *i* and *j* represent the spatial position information, *c* represents the output channel index of *F*$\left(x_{l-1}\right)$ and $N^{sp}\left(\cdot \right)$ uses the softmax function to emphasize the important areas in each channel. The attention mask is $N^{sp}\left(F\left(x_{l-1}\right)\right)$. The calculation method of the attention feature map *W* is defined as:(2)$$W\;=\;N^{sp}\left(F\left(x_{l-1}\right)\right)\cdot G\left(x_{l-1}\right)$$

The entire convolutional block learning is a combination of residual learning and attention mechanism learning. That is, the identity mapping is added at the element level.

Finally, the residual feature mapping of densely connected networks and the feature mapping of the attention mechanism are defined as follows:(3)$$x_{l}\;=\;F\left(x_{l-1}\right)+G\left(x_{l-1}\right)+bW$$
where *b* is a hyperparameter to trade-off between the feature map of the attention mechanism and the other two feature maps and *W* is the output of attention learning.

### 2.2. Loss Function

In the model training process, two binary classifiers are trained to discriminate between melanoma and others and between seborrheic keratosis and others. Since both models solve the problem of binary classification in the training process, this paper does not use the cross-entropy loss function of multi-classification. The cross-entropy loss function specifically for binary classification is defined as follows:(4)$$L_{loss}\;=\;-w_{i}[y_{i}{logx}_{n}+({1-y}_{i})log\left({1-x}_{i}\right)]$$
where $x_{i}$ represents the probability that the *i*-th lesion image is predicted as a positive example by the model, $y_{i}$ represents the real label of the *i*-th image and $w_{i}$ indicates the weight.

## 3. Experiments and Results

### 3.1. Implementation Details

Data preparation

The dataset used in this paper is hosted by the International Skin Imaging Collaboration (ISIC) 2016 and 2017. The International Skin Imaging Collaboration (ISIC) is an international effort to improve melanoma diagnosis, sponsored by the International Society for Digital Imaging of the Skin (ISDIS). The organization provides the largest publicly available dermoscopy image dataset of skin lesions.

The proposed model was trained and evaluated using two different datasets: two types of ISIC 2016 and three types of International Standard Industrial Classification 2017, as shown in Figure 4 and Figure 5 correspondingly.

ISIC 2016 consists of 900 training images and 379 test images. The lesion images are divided into two categories, malignant and benign.

ISIC 2017 provides 2000 training images, 150 validation images and 600 test images. The number of various types of lesions in the training set, validation set and test set is shown in Table 2. The lesion images break down into three categories: melanoma, nevus and seborrheic keratosis. Melanoma is commonly known as malignant skin cancer; it is derived from melanocytes. Nevus is benign skin cancer, derived from melanocytes, and seborrheic keratosis is benign skin cancer, derived from keratinocytes (non-melanocytes). There are two binary classification subtasks on this dataset: the classification of melanoma (melanoma vs. others) and the classification of seborrheic keratosis (seborrheic keratosis vs. others).

As presented in Table 2, there is a severe class imbalance in the dataset. Therefore, we adopt an oversampling method to address this problem. By this method, we generate samples of the minority class. We repeat the samples of each class in the training set and randomly sample from the samples of the minority class to add new samples so that all classes have an equal number of samples for training. We randomly select patches of different scales (1/5, 2/5, 3/5 and 4/5 of the original image) from the center of each image in the training set and then adjust to 224 × 224 by using bilinear interpolation. Each size operation generated 15 pictures, and each training image generated a total of 60 pictures. For example, the ISIC 2017 training set has a total of 2000 images, so the training data set was expanded by 60 times, with a total of 12,000 pictures. Data augmentation includes random cropping, random rotation and flipping.

The training set contains skin lesion images in JPEG format and CSV files. The CSV file contains some clinical metadata for each image. The first column of each row contains a string in the form of ISIC_<image_id>, where <image_id> matches the corresponding training data image. The second column in each row belongs to the first task of classification: malignant (melanoma) vs. benign (seborrheic keratosis/nevus), containing 0 (the lesion is melanoma) or 1 (the lesion is nevus or seborrheic keratosis). In ISIC 2017, the third column in each row belongs to the second classification task: melanocytes (melanoma) vs. non-melanocytes (seborrheic keratosis/nevus), containing 0 (seborrheic keratosis) or 1 (melanoma/nevus).

Network training

During the training process, the batch size was set to 10, the initial learning rate was 0.0001 and the maximum epoch was set to 100. The learning rate was reduced by half every 30 epochs. The network was adopted by the SGD as the optimizer, the attenuation of the weights was set to 0.0001 and the momentum was set to 0.9. The initial weighting factor for attention learning was set to 0.001. Then the official validation set was used to verify the performance of the model. In the test phase, we also used the official test set and selected a patch from each test image in the same way as in the training process and input these into the trained network to obtain the prediction score of the image. The unknown skin lesion dermoscopy image was fed to the trained model to classify the type of lesions.

The aim of this paper is to implement our model based on densely connected network attention and residual learning for skin lesion classification in Python under Pytorch.

Metrics

To evaluate the performance, we use average precision (AP), classification accuracy (ACC), sensitivity (SE), specificity (SP) and the area under the ROC curve (AUC) as the metrics in this paper.
(5)$$ACC\;=\;\frac{TP+TN}{TP+FN+TN+FP}$$
(6)$$Sensitivity\;=\;\frac{TP}{TP+FN}$$
(7)$$Specificity\;=\;\frac{TN}{TN+FP}$$
where:*TP* is true positive. It is judged as a positive sample; in fact it is also a positive sample.*TN* is true negative. It is judged as a negative sample; in fact it is also a negative sample.*FN* is false negative. It is judged as a negative sample, but in fact it is a positive sample.*FP* is false positive. It is judged as a positive sample, but in fact it is a negative sample.

### 3.2. Results

The proposed method is compared with several baseline models for classification in ISIC 2016 and ISIC 2017, referring to Table 3. (1) VGGNet has 16 layers of convolution, and the size of the convolution kernel is 3 × 3. (2) GoogLeNet is a convolutional neural network with 21 layers, part of which is composed of inception modules. The network has only one layer of fully connected convolutional neural network. (3) ResNet network compares the 50-layer and 100-layer residual network models. (4) DenseNet is the backbone of our model.

This paper compares our model with these methods, illustrated in Table 4 and Table 5.

#### 3.2.1. Compared with VGGNet, GooLeNet, ResNet and DenseNet in ISIC 2016

It can be seen from Table 3, VGGNet [13], GoogLeNet [14], ResNet [11] and DenseNet [16] that the depth of the convolutional neural network model is gradually increasing. The initial 16 layers gradually increased to hundreds of layers. Table 3 shows the number of convolution layers, the number of fully connected layers, the size of the convolution kernel and the size of the fully connected layers for each model.

Table 4 presents a comparison of model performance by applying VGGNet, GoogLeNet, ResNet50, ResNet101 and DenseNet100 models to the ISIC 2016 test set. The skin lesions on 379 test images are evaluated through SE, SP, ACC and AUC.

The classification performance can be seen so that when the network is deeper, the performance of the model becomes better across all metrics. It also shows that the proposed ARDT-DenseNet model achieves the best performance in skin lesion classification over other methods.

#### 3.2.2. Compared with VGGNet, GooLeNet, ResNet and DenseNet in ISIC 2017

These models are in Task 1 (malignant (melanoma) vs. benign (seborrheic keratosis/nevus)) and Task 2 (melanoma/nevus (melanocytes) vs. non-melanocytes (seborrheic keratosis)) in ISIC 2017. It can be seen from Table 5 that the number of network layers increases, and the accuracy of each model also gradually increases. The accuracy of the original 16-layer VGGNet model in Tasks 1 and 2 is 75.7% and 74.6%, respectively. Once the number of layers increased to 22, the accuracy of GoogLeNet is 76.4% and 75.8%, respectively. When the number of network layers is increased to 50, the accuracy rate of ResNet50 is increased by 7.1% and 8.0%, respectively.

It can be seen that the performance of the network model grows better with the increase of the number of convolution layers. This paper further studies that, when the number of network layers reaches the scale of hundreds, the accuracy rate of ResNet101 in Tasks 1 and 2 is 83.9% and 84.3%, respectively, and the accuracy of DenseNet100 reaches 84.2% and 85.0%, respectively. Compared with the 50-layer residual network, the 200-layer network models have greatly improved accuracy.

In addition, we also compared the number of parameters of these models. The number of parameters of the proposed VGGNet model reaches at most 138 × 10^6^, but the accuracy rate is only 75.7%to 74.6%. The optimized GoogLeNet not only improves its accuracy to 76.4% to 75.8% but also reduces the model parameters by about 20 times. Although the number of ResNet50 convolutional layers has increased, the number of parameters is only 25.5 × 10^6^, and the accuracy rate has reached 83.5%. ResNet101 not only has hundreds of layers but also has relatively large improvement in AUC, ACC, SE and SP over ResNet50. However, the number of parameters is twice that of ResNet50.

However, the DenseNet100 network model, which also has hundreds of layers, not only compares with ResNet101 in each evaluation index but also has only 21.2 × 10^6^ parameters, which is lower than the number of parameters of ResNet50 (25.5 × 10^6^). However, the number of convolutional layers is twice that of ResNet50. It can be seen that DenseNet100 has achieved relatively competitive results when the model sizes are not that different. Therefore, we chose DenseNet100 as the backbone network model.

In this paper, after determining the reference network model as DenseNet100, the method for attention and residual learning is introduced into the reference network model, called ARDT-DenseNet. The results of the evaluation in the two classifications are shown in Table 5. It is worth noting that the changes in the transition block not only do not introduce additional layers but also reduce the number of parameters of our model. After introducing attention and residual learning, the number of model parameters is still 20.3 × 10^6^. It can be clearly seen that, after the introduction of the new method, the ACC values of ARDT-DenseNet in Task 1 and Task 2 are 86.8% and 87.9%, respectively, which are 2.6% and 1.8% higher than for DenseNet100, respectively. The AUC values of ARDT-DenseNet in Task 1 and Task 2 are 87.9% and 95.7%, respectively, which are higher than those of DenseNet100.

In Task 1, ARDT-DenseNet was used to identify whether the skin of the lesion was melanoma (malignant) or seborrheic keratosis/nevus (benign). The value of AUC in Task 1 was 87.9%, which was 1.8% higher than that of DenseNet100. As shown in Figure 6, the red and black lines represent the result curves of ARDT-DenseNet and DenseNet100, respectively. In Task 2, the models are used to identify whether the skin lesion is non-melanocytes (seborrheic keratosis) or melanocytes (melanoma/nevus). The values of AUC in Task 2 are 93.9% and 95.7%, respectively. ARDT-DenseNet is improved over DenseNet100 by 1.8%. As shown in Figure 7, the pink and black lines represent the result curves of ARDT-DenseNet and DenseNet100, respectively.

The ARDT-DenseNet network model has made progress in both tasks of skin lesion classification, but the number of parameters has not increased relative to DenseNet100. This is because the attention learning introduced in ARDT-DenseNet does not introduce additional convolutional layers.

#### 3.2.3. ARDT-DenseNet Compared with State-of-the-Art Models and Top-Ranking Challenge Models in ISIC 2017

Our model is also compared with advanced skin lesion classification methods. Table 6 presents the performance comparison of the models by applying the advanced model ARL-CNN [31] and the top models of the challenge rankings in the ISIC-17 competition leaderboard of the ISIC 2017 test set. The performance of the dermoscopy images is evaluated through the SE, SP, ACC and AUC of the classification performance for skin lesions.

It is worth noting that the proposed ARDT-DenseNet model without external data obtained the highest mean AUC and the highest SE, ACC and AUC values in Task 1, and, in Task 2, it obtained the highest value for SE. For Reference [34] and Reference [35], although they have a high SP value in Task 1, indicating a high probability of not being misdiagnosed, their SE is 10.3% and 43.6%, respectively, which means low sensitivity and that, when diagnosing disease, the probability of not being missed is low. For a single metric, if the sensitivity of the diagnosis is increased, the specificity of the diagnosis will inevitably decrease. In other words, reducing missed diagnoses will inevitably increase misdiagnoses. Our method can achieve a good balance between the two metrics.

## 4. Discussion and Conclusions

Our model is also compared with advanced ensemble models of skin lesion classification, as shown in Table 7. As a matter of fact, these models cannot be compared directly with our model because of the ensemble strategy. However, to compare the state-of-the art methods with advanced performance and enough comparison information, we provide the ensemble models as a reference in Table 7. It can be seen that the ensemble model has superior performance on the skin lesion classification task. However, we can see that the SE of Task 1 in Reference [36] is 37.6%, which means that, when diagnosing disease, the probability of not being missed is low. Moreover, the ensemble model has a complex structure, which is usually difficult to train and time-consuming and requires more resources.

This paper studies the classification of skin lesions. Due to the high degree of similarity between different levels of skin lesion types, it has brought certain challenges to the classification of skin lesions. The deep convolutional neural network significantly improved the classification ability of the model, but it also brings a dramatic increment in the number of model parameters, introduces the computation cost, makes the model more difficult to train and therefore affects the result. For the densely connected network with 100 layers proposed in this paper, the number of parameters is greatly reduced, whereases the accuracy of skin lesion classification is improved compared with a general network with the same number of layers. The introduction of attention and residual learning not only avoids additional parameters but also strengthens the deep supervision of the network and improves the networks ability to identify the interested subareas of the image. In this paper, two classification tasks are separately tested on the dataset. Compared with other methods, the ARDT-DenseNet proposed in this paper achieved state-of-the-art results for skin lesion classification in the ISIC 2016 and ISIC 2017 datasets.

## 5. Future Work

Although the method has made some progress in the task of automatic classification of skin lesions in dermoscopy images, there is still space for improvement in research work. The problems in the classification of skin lesion images need to be further improved through research. For example, in the new ISIC 2018 dataset, lesion images are divided into seven categories in more detail, which requires further research on multi-classification issues. In future research, we will pay more attention to the most advanced classification models in deep learning and develop higher-performance network structures. In addition, we will explore whether the model proposed in this paper can be applied to other types of skin lesion recognition and to more fields.

## Figures and Tables

**Figure 1 sensors-20-07080-f001:**
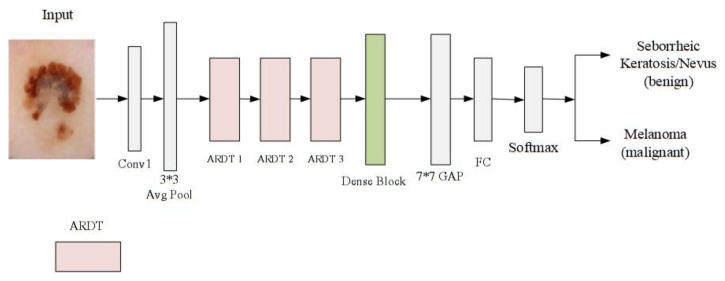
The architecture of the proposed network. The backbone network is DenseNet100. Dense blocks and transition blocks are described in Figure 2. ARDT blocks as shown in Figure 3.

**Figure 2 sensors-20-07080-f002:**
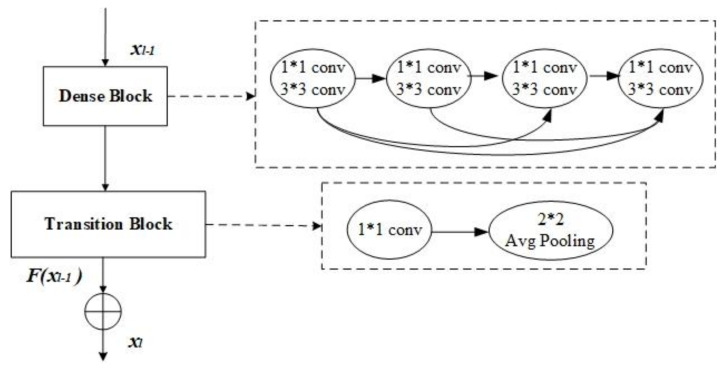
Architecture of the dense blocks and transition blocks. The “⊕” represents element-wise addition.

**Figure 4 sensors-20-07080-f004:**
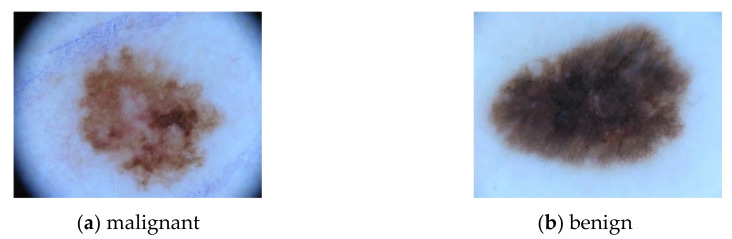
The International Skin Imaging Collaboration (ISIC) 2016 dataset contains two types of skin lesions: malignant (melanoma) (**a**) and benign (**b**). The lesions in these images are matched with the gold standard (definitive) malignant diagnosis, including 900 training images and 379 test dermoscopy images, which have been screened to ensure privacy and quality.

**Figure 5 sensors-20-07080-f005:**
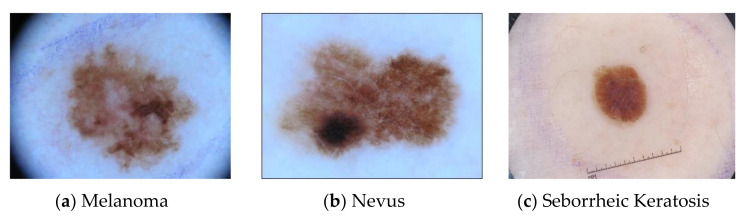
The lesion images in the ISIC 2017 dataset of the International Skin Imaging Collaboration are divided into three categories: melanoma (**a**), nevus (**b**) and seborrheic keratosis (**c**). This includes 2000 training, 150 verification and 600 test dermoscopy lesion images, which are all paired with gold standard diagnoses and ensure privacy and quality.

**Figure 6 sensors-20-07080-f006:**
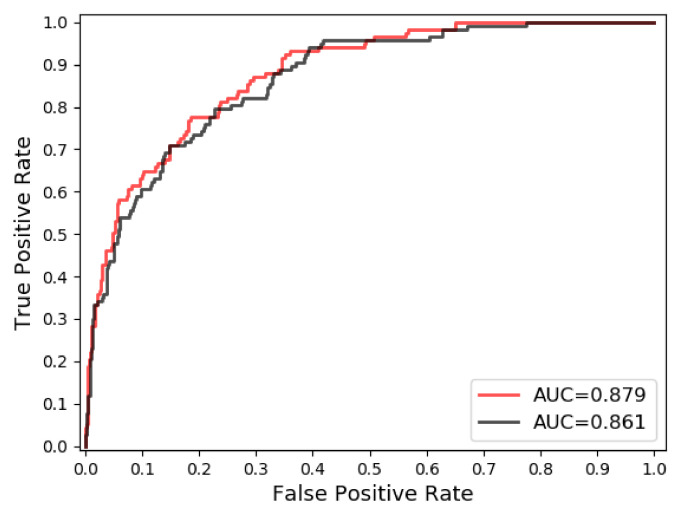
The red and black lines represent the result curves of ARDT-DenseNet100 and DenseNet100, respectively.

**Figure 7 sensors-20-07080-f007:**
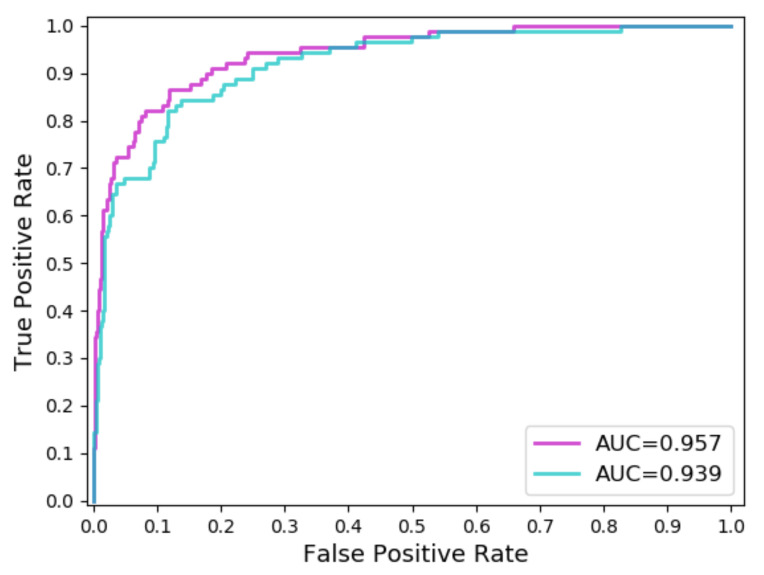
The pink and blue lines represent the result curves of ARDT-DenseNet and DenseNet100, respectively.

**Table 1 sensors-20-07080-t001:** The detailed structure of the DenseNet100 network.

Layers	Input Size	Output Size	Operations
Conv1	224 × 224	112 × 112	3 × 3 conv
Pooling	112 × 112	56 × 56	2 × 2 Avg pool
Dense Block 1	56 × 56	56 × 56	1 × 1 conv, 3 × 3 conv, 12
Transition layer 1	56 × 56	28 × 28	1 × 1 conv, 2 × 2 Avg pool
Dense Block 2	28 × 28	28 × 28	1 × 1 conv, 3 × 3 conv, 12
Transition layer 2	28 × 28	14 × 14	1 × 1 conv, 2 × 2 Avg pool
Dense Block 3	14 × 14	14 × 14	1 × 1 conv, 3 × 3 conv, 12
Transition layer 3	14 × 14	7 × 7	1 × 1 conv, 2 × 2 Avg pool
Dense Block 4	7 × 7	7 × 7	1 × 1 conv, 3 × 3 conv, 12
GAP	7 × 7	1 × 1	GAP
FC layer	1 × 1	1 × 1	FC
softmax			

**Table 2 sensors-20-07080-t002:** The number of various lesion types in the training set, validation set and test set.

	Melanoma	Seborrheic Keratosis	Nevus	Total
Training set	374	254	1372	2000
Validation set	30	42	78	150
Test set	117	90	393	600
Total	521	386	1843	2750

**Table 3 sensors-20-07080-t003:** The architecture of the baseline model.

	VGGNet	GoogLeNet	ResNet50	ResNet101	DenseNet100
Layer	16	22	50	101	100
Conv	13	21	49	100	99
FC	3	1	1	1	1
Convolutional kernel	3	7,2,3,5	7,1,3	7,1,3	4,4,4

**Table 4 sensors-20-07080-t004:** Comparison of baseline models on ISIC 2016.

Models	SE	SP	ACC	AUC
VGGNet	0.768	0.678	0.812	0.802
GoogLeNet	0.770	0.689	0.817	0.816
ResNet50	0.799	0.714	0.830	0.782
ResNet101	0.804	0.739	0.834	0.821
DenseNet100	0.812	0.742	0.842	0.817
ARDT-DenseNet	**0.816**	**0.756**	**0.857**	**0.837**

**Table 5 sensors-20-07080-t005:** Comparison of baseline models on ISIC 2017.

Models	Task 1				Task 2				$\mathrm{Params}\;({\times \;10}^{6})$
	SE	SP	ACC	AUC	SE	SP	ACC	AUC	
VGGNet	0.546	0.801	0.757	0.742	0.809	0.723	0.746	0.832	138
GoogLeNet	0.591	0.795	0.764	0.767	0.824	0.788	0.758	0.847	11.3
ResNet50	0.628	0.882	0.835	0.852	0.865	0.837	0.838	0.942	25.5
ResNet101	0.637	0.890	0.839	0.861	0.871	0.849	0.843	0.950	52.3
DenseNet100	0.641	0.887	0.842	0.861	0.869	0.852	0.850	0.939	21.2
ARDT-DenseNet	0.668	0.896	0.868	0.879	0.887	0.873	0.878	0.957	20.3

**Table 6 sensors-20-07080-t006:** ARDT-DenseNet compared with state-of-the-art models and top-ranking models in ISIC 2017.

Models	External Data	Ensembles	Task 1				Task 2				Mean AUC
			SE	SP	ACC	AUC	SE	SP	ACC	AUC	
ARL-CNN [31]	1320	N	0.658	0.896	0.850	0.875	0.878	0.867	0.868	0.958	0.917
#2 [34]	900	N	0.103	0.998	0.823	0.856	0.178	0.998	0.875	0.965	0.911
#6 [35]	0	N	0.436	0.925	0.830	0.830	0.700	0.995	0.917	0.942	0.886
ARL-CNN [31]	0	N	0.590	0.896	0.837	0.859	0.778	0.931	0.908	0.951	0.905
ARDT-DenseNet	0	N	0.668	0.896	0.868	0.879	0.887	0.873	0.878	0.957	0.918

**Table 7 sensors-20-07080-t007:** ARDT-DenseNet compared with state-of-the-art ensemble learning models in ISIC 2017.

Models	Ensembles	Task 1				Task 2				Mean AUC
		SE	SP	ACC	AUC	SE	SP	ACC	AUC	
Reference [36]	Y	**0.376**	**0.965**	0.850	0.891	0.722	**0.973**	0.935	0.960	0.926
Reference [37]	Y	-	-	-	0.892	-	-	**-**	0.966	0.929
ARDT-DenseNet	N	**0.668**	0.896	**0.868**	0.879	**0.887**	0.873	0.878	0.957	0.918

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
