# Peer review of "Skin Lesion Classification Using Densely Connected Convolutional Networks with Attention Residual Learning"

_sensors, 2020, doi:10.3390/s20247080_

Round 1
Reviewer 1 Report
The paper describes a method for skin lesion classification using densely connected convolutional blocks with attention mechanism. The paper is well-written in most parts and easy to follow.
Here are my comments about the paper to improve its quality:
- The literature review in the introduction part should be improved to cover recent works that use DL-based approaches for skin lesion classification. More specifically the following two works that explicitly use attention mechanism for skin lesion classification should be briefly explained.
- https://ieeexplore.ieee.org/document/8620285
- https://link.springer.com/chapter/10.1007/978-3-030-20351-1_62
- Specifically for the second aforementioned paper (Attention Residual Learning for Skin Lesion Classification), there are similarities in the methodological section (e.g. the utilized normalization method, similarities between ARL block and presented ARDT block, no additional parameters were added in the ARL block either). The authors should clearly explain the differences between their approach and other attention-based methods for skin lesion classification.
- What pre-processing steps were used? Especially were the images resized to a certain size before feeding them to the model?
- "This paper expands the training set by taking 60 different blocks for every image of the training set." What does it exactly mean? Does it mean that 60 cropped images were extracted from each input image? if yes, what was the crop size, and how the trained model was used in the inference phase?
- The reported results for ARL-CNN in Table 5 are not correct. Please modify them. For example, the average AUC is 91.7% in the actual paper but in this manuscript, an average AUC of 90.7% is reported.
- The comparison to the state of the art methods in table 5 is not complete. There are other published works with better performances in comparison to the ISIC 2017 challenge top performers. They should be added to table 5 (however authors can argue in the discussion section that the ensemble approaches or external datasets were used in other works). Papers that can be added to table 5 for comparison:
- https://arxiv.org/abs/2006.14715
- https://ieeexplore.ieee.org/document/9018274
- https://ieeexplore.ieee.org/document/8620285
- In figure 1, it seems there is no connection between the 3x3 average pooling layer and Dense block 1, but I assume it should be (as it can be also seen from figure 3)
- Figures quality should be enhanced, the texts in the figures are blurred to some extend.
Reviewer 2 Report
Skin Lesion Classification Using Densely Connected Convolutional Networks with Attention Residual Learning
Status: Major Revision
Check the correct writing in the English language of some sentences, verbs, prepositions, etc
Abstract:
- Both in the title and in the abstract, it is not explicitly stated that the work is focused on cancerous skin lesions, it is only mentioned at the beginning of the abstract in a small line and then only reference is made to "skin lesions" . It is important that from the beginning the problem that is attacking is promptly clear
- Introduction:
- 46: “diseases” → “disease”
- 49: “...it can observe” → “it can be observed”
- 58-61: There are works where the mentioned problems are reported?
- 65-66: “In general, by increasing the depth of the network, we can expect a better feature expression capability and an enhanced prediction performance.” → This is not always the case, it is not possible to generalize, mention situations and conditions where this is true
- 68: “...the recognition area in the image” → “...the image recognition area”
- 84: “...many methods adopt attention mechanism to...” → “..many methods that adopt attention mechanism to...”
- 101-111: In the introduction, contributions are usually mentioned in a general way, to be explained in detail in later sections. The way contributions are listed here may make some things not clear enough as they are not yet detailed in their corresponding section. Instead of this, a paragraph can be added explaining how the following sections are made up
- Materials and Methods
- 128: For a better visualization, the texts of the boxes in Figure 1 should be horizontal. The organization of this figure must be restructured
- 130-134: The mentioned parameters and their values ​​can be better described in a table to support the paragraph, and the choice of these values ​​must also be justified.
- 138-139: “The 1 × 1 convolutional in each transition block with 4 × 4 convolutional kernel, which facilitates further residual learning” → Restructure this statement to clarify it
- 140: “Because the purpose of the paper is to deal with the binary classification of skin lesions...” → The concept of binary classification has not been mentioned before, it must be made and mentioned because it is applicable to the problem of classification of skin lesions. This should be done in the introduction section
- 149-150: “To avoid this problem, we introduced the residual learning is shown in Figure 3...” → Restructure this statement
- 160-161: “The attention mechanism usually to learn attention weights by using additional trainable layers...” → Restructure this statement
- 184-186: This is precisely what I was asking in the comment on line 140. It should be mentioned before since it is a key aspect in the whole work to understand it.
Experiments and Results
- 219-223: Related to the previous comment. This has to be described earlier in the document, since they are key data to understand the subject and the way in which this proposal is presented.
- 224-228: Justify why this method was used for the dataset imbalance problem
- 232-239: These characteristics of the images should be mentioned when describing the dataset, a little above.
- 240-244: Why were these values ​​chosen? Are values ​​used in similar works? The proposal must be justified
- 245-246: “This paper is to implement skin lesion classification based on densely connected network attention and residual learning in Python under Pytorch.” → Restructure this statement
- 274: Acc or ACC?
- Very long chapter of experiments and results, it should be more punctual and some things should be passed to the conclusion and discussion section
Discussion and Conclusions
- 351: As I mentioned in the first comments, it should be clarified that they are carcinogenic skin lesions, since not all skin lesions are of this type
- 351-355: This is not a discussion and conclusions paragraph, it is only a summary
- 358: compared to which network? must be explicitly mentioned
364-371: This paragraph should be written in a future work section
Round 2
Reviewer 1 Report
I would like to thank the authors for their revision. My comments have been addressed in the revision and the paper quality has been improved.
As I mentioned in my comments before the proposed method has a large overlap with the introduced attention-based model in [30] (Attention Residual Learning for Skin Lesion Classification) although the number of parameters decreased from 23 million (in [30]) to 20.3 million in the proposed method. However, as the classification performance improved slightly (~1%), the proposed method can be beneficial for the research community in the field of skin lesion image analysis.
My only minor comment that was not addressed in the revision is the way of using the trained model in the inference phase that can be added to the manuscript.
Author Response
The authors are very grateful to the editor and the reviewers for their insightful comments and constructive suggestions. In light of these comments and suggestions, the paper has been very carefully revised.
I have improved the validation and teat stage in the first paragraph of “Network training”.
Thank you!
Reviewer 2 Report
Review of the comments response - Skin Lesion Classification Using Densely Connected Convolutional Networks with Attention Residual Learning
Point 1:
Ok, comments have been considered
Point 2:
Ok, comments have been considered
Point 3:
Ok, comments have been considered
Point 4:
Ok, comments have been considered
Point 5:
Ok, comments have been considered
Point 6:
Ok, comments have been considered
Point 7:
Ok, comments have been considered
Point 8:
Ok, comments have been considered
No more comments
Author Response
The authors are very grateful to the editor and the reviewers for their insightful comments and constructive suggestions, the paper has been very carefully revised. Thank you!